# Antagonistic Interaction between Phosphinothricin and *Nepeta rtanjensis* Essential Oil Affected Ammonium Metabolism and Antioxidant Defense of Arabidopsis Grown *In Vitro*

**DOI:** 10.3390/plants10010142

**Published:** 2021-01-12

**Authors:** Slavica Dmitrović, Milan Dragićević, Jelena Savić, Milica Milutinović, Suzana Živković, Vuk Maksimović, Dragana Matekalo, Mirjana Perišić, Danijela Mišić

**Affiliations:** 1Institute for Biological Research ‘‘Siniša Stanković”—National Institute of Republic of Serbia, University of Belgrade, Bulevar despota Stefana 142, 11060 Belgrade, Serbia; mdragicevic@ibiss.bg.ac.rs (M.D.); savic.jelena@ibiss.bg.ac.rs (J.S.); milica.milutinovic@ibiss.bg.ac.rs (M.M.); suzy@ibiss.bg.ac.rs (S.Ž.); dragana.bozic@ibiss.bg.ac.rs (D.M.); 2Institute for Multidisciplinary Research, University of Belgrade, Kneza Višeslava 1, 11030 Belgrade, Serbia; maxivuk@imsi.rs; 3Institute of Physics Belgrade—National Institute of the Republic of Serbia, University of Belgrade, Pregrevica 118, 11080 Belgrade, Serbia; mirjana@ipb.ac.rs

**Keywords:** BASTA, phosphinothricin, *Nepeta*, essential oil, glutamine synthetase, ammonium toxicity, antioxidant defense, Arabidopsis, antagonism

## Abstract

Phosphinothricin (PPT) is one of the most widely used herbicides. PTT targets glutamine synthetase (GS) activity in plants, and its phytotoxicity is ascribed to ammonium accumulation and reactive oxygen species bursts, which drives rapid lipid peroxidation of cell membranes. In agricultural fields, PPT is extensively sprayed on plant foliage; however, a portion of the herbicide reaches the soil. According to the present study, PPT absorbed via roots can be phytotoxic to Arabidopsis, inducing more adverse effects in roots than in shoots. Alterations in plant physiology caused by 10 days exposure to herbicide via roots are reflected through growth suppression, reduced chlorophyll content, perturbations in the sugar and organic acid metabolism, modifications in the activities and abundances of GS, catalase, peroxidase, and superoxide dismutase. Antagonistic interaction of *Nepeta rtanjensis* essential oil (*Nr*EO) and PPT, emphasizes the existence of complex control mechanisms at the transcriptional and posttranslational level, which result in the mitigation of PPT-induced ammonium toxicity and in providing more efficient antioxidant defense of plants. Simultaneous application of the two agents in the field cannot be recommended; however, *Nr*EO might be considered as the PPT post-treatment for reducing harmful effects of herbicide residues in the soil on non-target plants.

## 1. Introduction

Herbicides remain the primary tool for implementing weed management to maintain high yields of economically important crops. Phosphinothricin (PPT), also known as glufosinate (commonly used in the form of glufosinate-ammonium), is the major active ingredient in many non-selective herbicide formulations, including BASTA^®^ (BASF SE, Germany; previously Bayer Crop Science AG, Germany). It acts by inhibiting glutamine synthetase (GS) activity in plants [1,2,3,4,5], thus disabling the utilization of ammonium [2,6,7]. Accumulation of ammonium is highly phytotoxic, it provokes growth inhibition, leaf chlorosis and root atrophy [4,8,9,10,11], and induces oxidative stress, which is accompanied by the production of reactive oxygen species (ROS) [12,13] and increased activities of antioxidative enzymes [14]. Takano et al. [15] proposed that GS inhibition provokes the disruption of photorespiration and light reactions of photosynthesis, which leads to the photoreduction of molecular oxygen, and subsequent generation of ROS [16]. Actually, the equilibrium between the production and scavenging of ROS is disturbed when GS is inhibited [15]. Takano et al. [15,17] further suggested that the production of ROS, which drives the harmful lipid peroxidation of cell membranes and rapid cell death, is the major cause of rapid PPT toxicity. PPT provokes detrimental effects on the overall plant physiology and biochemistry, causing plant cell death within a few hours of treatment.

Under low concentrations of PPT, its uptake by plant leaves is probably driven by an active transporter, and it is suggested that PPT and glutamine may compete for the same transporter [15]. Under common field conditions, absorption mostly occurs by cell-to-cell diffusion due to high PPT concentrations [15]. After PPT is absorbed, it is translocated by the apoplast in the xylem, which is dependent on the transpiration rate [18]. During its foliar application under field conditions or when the affected leaves fall off on the soil before harvesting, a part of the herbicide reaches the soil. Phosphinothricin does not remain in the soil for long because it is rapidly degraded by the soil microorganisms via oxidation, transamination, and acetylation reactions [16]. According to estimations, the half-life of PPT and its residues varies from 1 to 25 days [16,19,20,21,22], depending on the type of soil and environmental conditions. In sterile soils, the half-life of PPT is longer. However, PPT is susceptible to leaching from the soil surface [19], but it is degraded before it reaches the lower soil layers and aquatic ecosystems [23]. As PPT is highly soluble in water it might also be absorbed by plants via roots [24]. The absorption and translocation mechanisms, as well as the mode of action of PTT entering the plant via roots is not well understood, but it could contribute to the overall toxicity of the herbicide. 

Mixtures containing PPT and other herbicides offer opportunities to improve the efficacy of PPT. For example, synergistic effects of PPT and 2,4-D or PTT and dicamba are well known [25,26,27]. However, in some cases, antagonistic effects were described. PPT efficacy was reduced in mixtures with monosodium methyl arsenate [28], or with the essential oil (EO) of *Nepeta rtanjensis* [4], which is an interesting bioherbicide for weeds. This EO, rich in iridoid monoterpenoids nepetalactones, was phytotoxic for *Ambrosia artemisifollia* [29], *Stellaria media* (L.) Vill, *Rumex crispus* L. [30] and *Arabidopsis thaliana* L. Heynh [4]. Joint foliar application of BASTA and *Nepeta rtanjensis* EO (*Nr*EO), resulted in the preservation of GS activity in Arabidopsis along with the maintenance of sub-toxic and/or sub-lethal ammonium concentrations in tissues [4]. In search for more explanations of the described phenomenon, we hypothesized that the PPT present in the Arabidopsis root surrounding also displays herbicidal effects, which can be mitigated by *Nr*EO. We further hypothesized that antagonistic interaction between the two agents involves not only GS activity preservation, but also the perturbations in the antioxidant system of Arabidopsis. In order to test the postulated hypotheses, we prepared an *in vitro* experimental setup that enabled PPT supply through the culture medium, and exposure of Arabidopsis to *Nr*EO volatiles via the atmosphere of culture vessels.

## 2. Results

The *in vitro* experimental setup enabled the 10-day exposure of Arabidopsis roots to BASTA (B5 and B10), while *Nr*EO (2*Nr*EO and 4*Nr*EO) components were present in the atmosphere within the glass vessels (Figure 1A). We examined the changes induced by BASTA and *Nr*EO in shoots and roots individually, in a dose-dependent manner. Organic volatiles present in the atmosphere of the glass vessels, released from the surface of the filter paper moistened with *Nr*EO (2*Nr*EO and 4*Nr*EO), were quantified using PTR-MS (Figure 1B). The amount of nepetalactones (*cis*,*trans*- and *trans*,*cis*-nepetalactone) with [M+1]^+^ at *m*/*z* 167, and of total monoterpenoids with the protonated masses at *m*/*z* 137 and *m*/*z* 153 were traced. Compounds with [M+1]^+^ at *m*/*z* 205 were also analyzed. All these compounds were previously identified in *Nr*EO by GC–MS and GC–FID analyses [29,31,32,33]. Concentrations of *Nr*EO (2% and 4%) at the beginning of the experiment resulted in ~1700 and ~3400 ppbV of nepetalactones in the atmosphere of glass vessels, respectively. However, the nepetalactone concentration in the atmosphere of the glass vessels was severely decreased after 10 days: the concentration of nepetalactones in treatments with 2*Nr*EO was around 18 ppbV, while these compounds reached 29 ppbV in treatments with 4*Nr*EO (Figure 1B). As expected, nepetalactone was not detected in the control group of plants nor in the treatments with BASTA alone (Figure 1B). The same trend was observed for other traced compounds. Interestingly, 10 days after the beginning of the experiment, nepetalactones (with the *m*/*z* 167) were the second most abundant group of analyzed terpenoids. More abundant were monoterpenoids that exhibited [M+1]^+^ at *m*/*z* 137 (Figure 1B), which might be, at least partially, the result of the accumulation of compounds with the same masses released from the surface of the Arabidopsis leaves.

### 2.1. Growth and Metabolism of Arabidopsis as Influenced by BASTA and/or NrEO

BASTA-exposed Arabidopsis was characterized by the highly significant reduced fresh weight (FW) of shoots and roots (Appendix A), and this decrease was dose-dependent in shoots (Figure 1C). However, FW of Arabidopsis shoots and roots were not significantly affected by *Nr*EO treatments, while the simultaneous application of the two agents (BASTA/*Nr*EO interaction) had a significant effect on shoot FW (Appendix A). Post-hoc analysis suggested significant growth reduction of Arabidopsis shoots by *Nr*EO, which was also dose-dependent and more pronounced at higher *Nr*EO concentrations (Figure 1C). Simultaneous application of the two agents did not induce additional FW reduction compared to separate treatments with each factor, and in the case of 4*Nr*EO+B10 treatment, a slight but not significant increase in shoot FW was observed compared to separate 4*Nr*EO and B10 treatments (Figure 1C). Chlorophyll (Chl) content was the highest in non-treated Arabidopsis shoots, where the Chl a+b concentration reached ~450 μg g^−1^ FW. Exposure of plants to BASTA via-roots resulted in visible discoloration of *in vitro* grown Arabidopsis shoots (Figure 1A), which was most likely accompanied by chlorophyll (Chl) degradation or its reduced synthesis. The reduction of Chl content was dose-dependent and more pronounced at higher BASTA concentrations, where around 125 μg g^−1^ FW of Chl a+b was recorded (Figure 1D). The observed BASTA effect was statistically significant (*p* < 0.001) (Appendix A). However, *N. rtanjensis* EO (both 2*Nr*EO and 4*Nr*EO treatments) had no significant effect on Chl content in Arabidopsis shoots (Appendix A). Simultaneous application of BASTA and *Nr*EO induced a discoloration effect similar to that observed in BASTA treatments. The exception was the treatment with 4*Nr*EO+B10, where plants exhibited a slightly lower reduction in Chl a, Chl b and Chl a+b content, in comparison to those treated with B10 alone. Non-treated Arabidopsis were characterized by the presence of high amounts of isocitric (~3.5 μg mg^−1^ FW) and fumaric acid (~2.9 μg mg^−1^ FW). Malic (~1.9 μg mg^−1^ FW), succinic (~1.6 μg mg^−1^ FW) and oxalic acid (traces) were present in significantly lower amounts. Our results revealed that BASTA treatment altered the amount and ratio of organic acids in Arabidopsis shoots. It induced the increment in the content of oxalic, malic and isocitric acid, while succinic and fumaric acid content in Arabidopsis shoots decreased (Figure 1D). These trends were especially obvious in B10 treatments. However, application of *Nr*EO significantly increased the content of isocitric, malic, and succinic acid. The contents of oxalic, isocitric, malic and fumaric acids in Arabidopsis shoots simultaneously treated with BASTA and *Nr*EO, followed the trends observed in treatments with BASTA alone. Succinic acid content in shoots increased significantly following joint B5 and *Nr*EO (both 2% and 4%) treatments, as compared to B5-treated plants. ANOVA indicated significant effects of BASTA on oxalic, isocitric, malic, succinic (*p* < 0.001) and fumaric acid (*p* < 0.05) content in Arabidopsis shoots, while *Nr*EO significantly affected succinic (*p* < 0.001), malic and isocitric acid (*p* < 0.05) content (Appendix A). Significant interaction (*p* < 0.01) of the two agents was only recorded for succinic acid (Appendix A). The main sugars identified in Arabidopsis shoots were sucrose (Suc, ~387 mg 100 mg^−1^ FW), glucose (Glu, ~138 mg 100 mg^−1^ FW) and fructose (Fru, ~126 mg 100 mg^−1^ FW). The content of Suc increased upon treatments with BASTA, in a dose-dependent manner (Figure 1D). At B10 treatments, the Suc concentration reached 911 mg 100 mg^−1^ FW. Nevertheless, application of *Nr*EO resulted in a slight decrease in Fru (~98 mg 100 mg^−1^ FW) and Suc (~300 mg 100 mg^−1^ FW) content in Arabidopsis shoots. As growth was prevented, carbohydrate accumulation in shoots could be attributed to the lack of their utilization. Joint action of BASTA and *Nr*EO induced a slight decrease in Fru and Suc content, when compared to BASTA-treated plants. ANOVA statistical analysis indicated the significant influence of BASTA and *Nr*EO on Fru and Suc content in Arabidopsis shoots, while the concomitant effect of the two agents was not significant (Appendix A).

### 2.2. Ammonium Metabolism in Arabidopsis Shoots and Roots as Influenced by BASTA and/or NrEO

The activity of GS was generally higher in Arabidopsis roots than in shoots (Figure 2A). While the application of BASTA significantly reduced GS activity in both shoots and roots, *Nr*EO showed no significant effect. Simultaneous application of BASTA and *Nr*EO slightly mitigated the inhibitory effect of BASTA on GS activity in Arabidopsis roots; but still, this effect was not statistically significant according to Tukey’s HSD post-hoc test, except for 2*Nr*EO+B10 treatment. In terms of GS activity, analyses of variance indicated a significant effect of BASTA in both shoots and roots, while *Nr*EO was only effective in roots (Appendix A). Native-PAGE zymograms of Arabidopsis proteins from shoots and roots (Figure 2B), stained for GS activity, displayed multiple GS activity bands. In shoots, the activities and mobility of individual isoforms varied in response to BASTA and *Nr*EO treatments. In the non-treated plants and in plants treated with *Nr*EO, GS isoforms in shoots with higher mobility were more active in comparison to low-mobility isoforms. Application of B5 and B10, and simultaneous application of BASTA and *Nr*EO increased the activity of some low-mobility isoforms in shoots (Figure 2B). BASTA increased the activity of highly mobile GS isoforms in roots, while *Nr*EO decreased their activity (Figure 2B). Immunoblotting analysis of GS proteins in shoots and roots of *in vitro* grown Arabidopsis (Figure 2C) revealed two distinct GS bands, which were assigned to GS1 (40 kDa) and GS2 (44 kDa) according to literature data [34]. GS1 proteins were proved to be more abundant than GS2 proteins in both shoots and roots. However, GS1 proteins were more abundant in roots, than in shoots. Following BASTA treatments, the amount of GS proteins in shoots and roots increased, and this increment was more pronounced at higher BASTA concentrations (B10) in roots (Figure 2B). The application of *Nr*EO decreased the amount of GS1 proteins in Arabidopsis roots, while in shoots their amount was unchanged compared to already low amounts in the control. Interestingly, simultaneous application of *Nr*EO and BASTA in shoots, led to increased amount of GS1 proteins compared to non-treated plants; however, this increment was lower than in treatments with BASTA only, with the exception for 2*Nr*EO+B10 treatment. The results indicated an obvious suppression of BASTA-induced changes in GS1 abundance. This phenomenon was evident in both shoots and roots. Abundance of GS2 was also slightly increased in roots upon BASTA treatments, but it was not affected when BASTA was applied simultaneously with *Nr*EO (Figure 2C). 

The reduction in GS activity in BASTA-treated Arabidopsis shoots and roots were accompanied by a significant increase in ammonium accumulation (Figure 2A,D, Appendix A). Volatile organic compounds (VOCs) of *Nr*EO present in the glass vessels showed no effect on ammonium accumulation in shoots and roots, while the BASTA/*Nr*EO interaction was significant (*p* < 0.05) in shoots (Appendix A). The effect of the interaction can be observed since the simultaneous application of B5 and *Nr*EO resulted in a reduced ammonium accumulation in shoots, when compared to B5-treated Arabidopsis (Figure 2D), but this reduction was not statistically significant according to Tukey’s HSD test. 

Expression of *GS1* and *GS2* genes in Arabidopsis shoots and roots was analyzed. *In vitro* grown Arabidopsis accumulated different levels of *GS1* and *GS2* gene transcripts, i.e., transcripts of *GS1* genes were more abundant than those of *GS2*, in both *in vitro*-grown shoots and roots (Appendix A). Among the *GS1* genes, transcripts of *GLN1;2* were the most abundant in shoots and roots, and were followed by *GLN1;3*, *GLN1;1*, and *GLN1;4*. The expression of plastid *GLN2* isoform was low in both shoots and roots (Appendix A).

The expression patterns of all analyzed GS genes in Arabidopsis treated with BASTA, *Nr*EO, *Nr*EO+BASTA, and in the control group of plants, displayed rather opposite trends in shoots and roots (Figure 3). Both BASTA treatments (B5 and B10) induced a slight increase in *GLN1;2*, *GLN1;3*, and *GLN1;4* expressions in shoots, while the expression of *GLN2* was slightly decreased. Treatment with B10 significantly enhanced the expression of *GLN1;1* in shoots. Changes observed for *GS1* genes were dose-dependent, and were particularly pronounced upon B10 treatments (Figure 3). However, BASTA treatment clearly down-regulated all five tested genes in roots (Figure 3), with *GLN1;1*, *GLN1;2* and *GLN1;3* being significantly affected at higher BASTA concentrations (B10). The application of 2*Nr*EO or 4*Nr*EO induced no significant changes in the expression of *GLN1;1*, *GLN1;2*, *GLN1;3*, *GLN1;4*, and *GLN2* in shoots of Arabidopsis, while in roots it resulted in slightly decreased expressions of all five genes; these changes were statistically significant for *GLN1;1* and *GLN1;3* at 2*Nr*EO treatment (Figure 3). Simultaneous treatment with BASTA and *Nr*EO followed the trends observed in BASTA treatments for all analyzed genes in Arabidopsis shoots. However, simultaneous application of *Nr*EO (especially of 4*Nr*EO) with BASTA, reduced the inhibitory effect of BASTA on the expression of *GLN1;1*, *GLN1;2*, *GLN1;3*, *GLN1;4* and *GLN2* in roots (Figure 3, Appendix A). Therefore, BASTA and *Nr*EO interaction was not statistically significant for the expression of five analyzed genes in Arabidopsis shoots, while in roots it significantly affected the expression of all GS genes according to the analysis of variance (Appendix A).

### 2.3. Antioxidant Enzymes Activity in Arabidopsis Shoots and Roots as Influenced by BASTA and/or NrEO

The activities and abundance of antioxidant enzymes, catalase (CAT), peroxidase (POX) and superoxide dismutase (SOD), were used as biochemical markers of the stress resistance capacity of *in vitro* grown Arabidopsis to BASTA (B5 and B10) and/or *Nepeta rtanjensis* essential oil (2*Nr*EO and 4*Nr*EO).

Three CAT isoforms were detected in Arabidopsis shoot and root samples (Figure 4A,B). A strong decrease of CAT activity was recorded in shoots and roots of Arabidopsis treated with *Nr*EO (2*Nr*EO and 4*Nr*EO) for 10 days, when compared with non-treated plants (nt) (Figure 4A,B). Treatment with *Nr*EO reduced the activity of all three CAT isoforms, and this effect was more pronounced in shoots (Figure 4A) than in roots (Figure 4B). BASTA treatments (B5 and B10) only induced a slight decrease in CAT activity in shoots (Figure 4A). In the roots the activity of CAT was increased (Figure 4B), probably because of the induced activity of the low mobility CAT1 isoform, which is the major contributor to the overall activity of this enzyme. Joint application of BASTA and *Nr*EO resulted in the decline of CAT activity in shoots, with the exception of 4*Nr*EO+B5 treatment, which induced no changes. Increase in the activity of high mobility CAT3 isoform and decrease in the activity of CAT1 were recorded on B10 treatments with 2*Nr*EO and 4NrEO (Figure 4A), when compared to the B10 treatment. In roots, both *Nr*EO+B10 treatments increased the CAT activity, especially of CAT1 isoform, when compared to the non-treated plants (Figure 4B). Changes in the activities of CATs in Arabidopsis shoots and roots influenced by BASTA and/or *Nr*EO treatments, are followed by changes in the abundance of this enzyme in plant organs (Figure 4C,D). Treatments with BASTA increased the abundance of CATs in roots, while *Nr*EO decreased the abundance of this enzyme in both organs. Abundance of CAT in Arabidopsis shoots and roots exposed to *Nr*EO+B5 action was decreased when compared to B5-treated plants. In roots, on treatments with B10 and 2*Nr*EO or 4*Nr*EO, the abundance of enzyme was increased, when compared to B10 treatment (Figure 4D).

Native-PAGE gels stained for POX activity, after 10 days of BASTA and *Nr*EO treatments, revealed changes in Arabidopsis shoots and roots. In shoots, POX activity (especially that of the high mobility POX3 isoform) was increased by the application of B10, alone or in combination with 2*Nr*EO or 4*Nr*EO (Figure 5A). When applied separately, 2*Nr*EO induced around a 20% decrease in POX activity in shoots. In roots, B5 and B10 treatments induced significant increase in total POX activity, while 2*Nr*EO and 4*Nr*EO treatments reduced the activity of this enzyme (Figure 5B). Both POX1 and POX2 isoforms were equally affected. Exposure of Arabidopsis to 2*Nr*EO+B5 and 4*Nr*EO+B5 treatments decreased the POX activity in roots, when compared to B5 treatment. Treatment with 4*Nr*EO+B10 slightly decreased the activity of the enzyme, when compared to B10-treated plants (Figure 5B). Treatments with 2*Nr*EO+B10 and 4*Nr*EO+B10 significantly increased POX activity, when compared to the non-treated plants (Figure 5B). Based on immunoblot analysis eight isoforms in shoots (POXs) and seven in roots (POXr) can be distinguished (Figure 5C,D). In shoots, the most abundant isoforms were POX4s and POX6s, while in roots POX6r, POX7r and POX3r prevailed. The highest total POX abundance in shoots was recorded on treatments with BASTA (B5 and B10), while the 2*Nr*EO treatment was the most efficient in reducing the abundance of this enzyme (Figure 5C). This is especially visible for the most abundant POX isoforms in shoots (POX4s and POX6s). It was demonstrated that application of *Nr*EO (both 2*Nr*EO and 4*Nr*EO) simultaneously with BASTA (B5 and B10) reduced the abundance of POX isoforms, in comparison to appropriate controls (B5 and B10, respectively) (Figure 5C). In roots, on the other hand, BASTA treatments reduced the abundance of the majority of POX isoforms, while *Nr*EO treatments increased the abundance of this enzyme (Figure 5D). The exception is the abundance of POX7r isoform, which was increased following B10 application. Joint application of BASTA and *Nr*EO generally reduced the abundance of POX isoforms in roots when compared to appropriate BASTA treatments.

Native-PAGE in-gels assays for SOD activity revealed that Arabidopsis exposed to BASTA (B5 and B10) and/or *Nr*EO (2*Nr*EO and 4*Nr*EO) treatments for 10 days, demonstrated unchanged activity in all shoot samples (Figure 6A), whereas the SOD activity was decreased following 2*Nr*EO and 2*Nr*EO+B5 treatments in roots (Figure 6E). Three SOD isoforms were detected in shoot samples, and five in roots. Based on different sensitivity to corresponding inhibitors, Mn-, Fe-, and CuZn-SOD isoforms were demonstrated in Arabidopsis shoots and roots (Figure 6B,F). In Arabidopsis shoots, CuZn-SOD isoform was the major contributor to the overall SOD activity, and was followed by Mn-SOD and Fe-SOD isoforms. In roots, one Fe-SOD and one CuZn-SOD were recorded, as well as three Mn-SOD isoforms (Figure 6E,F). The abundance of Mn-SOD and Fe-SOD isoforms in Arabidopsis shoots and roots was evaluated using immunodetection assay (Figure 6C,D,G,H). All treatments, except 2*Nr*EO, induced a decrease in Mn-SOD abundance in shoots, in comparison to non-treated plants (Figure 6C). The opposite model of action was observed in root samples, since all treatments, except 2*Nr*EO and 2*Nr*EO+B5, increased Mn-SOD abundance in comparison to non-treated plants (Figure 6G). The increase was the most pronounced on B5 treatment. Interestingly, the application of *Nr*EO (2*Nr*EO and 4*Nr*EO) simultaneously with BASTA (B5 and B10) in shoots mitigated the inhibitory effects of herbicide on Mn-SOD abundance (Figure 6C). In roots, *Nr*EO mitigated the stimulatory effect of B5 on Mn-SOD abundance (Figure 6G). As for the Fe-SOD isoform abundance, it was increased on all treatments, with the exception of B10 treatment, where changes were not observed (Figure 6D). In roots, a similar trend was observed as in shoots (Figure 6H), and FeSOD abundance was especially increased on B5 treatment. The results of immunodetection assay revealed significant amounts of Fe-SOD protein in both shoots and roots (Figure 6D,H).

## 3. Discussion

Phosphinothricin can be absorbed via roots, and further efficiently transported through the xylem sap to the above-ground parts of plants [24]. Information regarding PPT root uptake are scarce, therefore our intention was to describe the effects of PPT absorbed via roots on physiology and metabolism of Arabidopsis, and thus to get an insight into its contribution to the overall herbicidal action. *In vitro* experimental setup enabled the direct exposure of roots to BASTA incorporated in the sterile culture medium, where metabolizing by microorganisms is excluded. We further aimed to investigate whether the phenomenon of *Nr*EO-mediated mitigation of the BASTA-induced ammonium toxicity in Arabidopsis, observed during their simultaneous foliar application [4], can also be achieved when plants are exposed to herbicide via roots. It was possible to expose Arabidopsis to the atmosphere enriched with volatiles of *N. rtanjensis* EO, and to trace their amount by appropriate contemporary methodology, such as PTR-MS. Nepetalactone amount in the atmosphere of *in vitro* culture vessels was obviously dependent on the applied concentration of *Nr*EO, and it decreased gradually over 10 days of in vitro cultivation. As previously suggested by Dmitrović et al. [29], this decrease might result from VOCs transitions between liquid and gas phases, and their degradation, transformation, precipitation, or leakage from the glass vessels to the ambient air. Either way, the amount of this monoterpenoid was sufficient to induce nepetalactone-expected effects on Arabidopsis, when applied alone or in combination with BASTA. 

Apparent BASTA-induced effects, similar to those induced by foliar application of herbicide [4], were visible within the present experimental setup, which indicated efficient PPT translocation from roots to the above ground plant parts. Arabidopsis displayed clear ammonium toxicity symptoms in the form of leaf chlorosis and reduced plant growth, accompanied by decreased GS activity and increased tissue ammonium content in shoots and roots. Li et al. [11] demonstrated that shoot tissues of Arabidopsis are hypersensitive to ammonium exposure, compared to roots that store ammonium in vacuoles, thus avoiding ammonium toxicity. This is in accordance with our findings that the reduction in GS activity was more pronounced in Arabidopsis roots, where a less pronounced increase in ammonium content was recorded when compared to the shoots. The most severe BASTA-phenotype was observed following B10 treatments. Leaf chlorosis was accompanied by a decline in Chl a, Chl b and Chl a+b content, as well as by perturbation in the content of other traced metabolites (soluble sugars and organic acids). Treatments with *Nr*EO, however, induced expected changes in Arabidopsis growth, chlorophyll content, sugars and organic acid content. As in our previous study that adopted foliar *Nr*EO application [4], Arabidopsis shoots exposed to 2*Nr*EO and 4*Nr*EO via atmosphere displayed growth reduction and a decrease in soluble sugars content, but no significant changes in organic acids content. Applied in a similar way, this EO has previously been reported to induce growth inhibition of ragweed (*Ambrosia artemisiifolia* L.) shoots [29], and garden cress seedlings (*Lepidium sativum*) [35]. Regardless of the method of application (foliar or via-atmosphere), *Nr*EO induced a decrease in Chl a, Chl b and Chl a+b content in Arabidopsis [4], but also in ragweed shoots [29]. The result of BASTA and *Nr*EO joint application was the reduced growth of Arabidopsis shoots when compared to the non-treated plants, while roots were not affected. Generally, the recorded values of changes in chlorophyll and organic acids content in Arabidopsis shoots following *Nr*EO+BASTA treatments, are the mean values of the two agents applied independently. 

As the PPT mechanism of action is based on its inhibitory effects on GS activity, we traced the effects of BASTA on GS activity and GS-coding genes expression, as well as on the content of ammonium in Arabidopsis shoots and roots. There are two classes of GS isoforms in plants: cytosolic GS1 and plastidic GS2. GS1 isoforms are located in the cytoplasm and are encoded by a multigene family (*GLN1;1*, *GLN1;2*, *GLN1;3*, *GLN1;4* and *GLN1;5*)*,* while GS2 is encoded by the *GLN2* gene in Arabidopsis [34,36]. Opposite to the greenhouse-grown Arabidopsis, where *GLN2* expression in leaves was higher compared do *GS1* genes [4,37], transcript levels of *GS1* (especially *GLN1;2*) were more abundant compared to *GLN2* in plants grown under in vitro conditions. This suggested that GS1 is the major isoform responsible for ammonium assimilation in both shoots and roots of Arabidopsis grown in vitro. Taking into account in vitro mixotrophic growth conditions, which include external carbohydrate supply through culture medium and lowered photosynthetic rate, it was not surprising that the expression of plastidic *GLN2* isoform was low in both shoots and roots. However, it is well documented that expression levels of *GLN1;1* and *GLN1;2* in Arabidopsis are modulated in response to abiotic stresses, including ammonium toxicity [4,37], low and/or high N [38], and salt stress [39]. Ammonium accumulation induced by BASTA treatment within the present study induced the expression of *GS1* (*GLN1;1, GLN1;2, GLN1;3*, and *GLN1;4*) in Arabidopsis shoots, which was positively correlated with the amount of GS1 proteins, but negatively correlated with the activity of GS. This indicated, on the one hand, the existence of transcriptional regulation of GS expression by BASTA, probably via accumulated ammonium, but also the potent inhibition of GS activity by the herbicide. Such a mode of action was also recorded in Arabidopsis leaves during foliar application of herbicide [4]. As previously suggested, PPT irreversibly, but not covalently, binds to the active site of GS and inhibits its activity [1,40,41]. The result of reduced GS activity is the accumulation of phytotoxic ammonium in tissues, which alters the overall physiology of Arabidopsis. The activity and abundance of GS are generally higher in Arabidopsis roots, than in shoots, and BASTA-induced reduction in GS activity was more pronounced in roots. However, Arabidopsis shoots experienced more extensive ammonia accumulation than roots. Roots have efficient mechanisms for ammonium detoxification, which, as previously suggested [11], might include its storage in vacuoles.

The application of *NrEO* induced no significant changes in the expression of *GLN1;1*, *GLN1;2*, *GLN1;3*, *GLN1;4*, and *GLN2* in Arabidopsis shoots, on GS1 and GS2 abundance and activity, as well as on ammonium content. This is in accordance with our previous study, when foliar application of *Nr*EO was adopted [4]. Furthermore, no obvious differences between BASTA and *Nr*EO+BASTA treated plants were visible at the level of *GS1* and *GS2* gene expression in shoots. However, GS abundance in shoots was higher in BASTA treatments than in *Nr*EO+BASTA treatments, with the exception of 2*Nr*EO+B10 treatment. The reason for such a discrepancy might be the induced expression of *GLN* genes earlier during the treatment which results in the increased amount of GS proteins and subsequent reduction of gene expression to the control values, which is visible after 10 days of treatments. This presumption is in accordance with our previous study, which revealed that changes in GS genes expression are more pronounced 1 day after the beginning of BASTA and/or *Nr*EO treatments than after 10 days [4]. In roots, the situation was even more complex. ANOVA indicated that the exposure of plants to *Nr*EO induced significant down-regulation of all analyzed genes. Post-hoc analysis indicated that *GLN1;1* and *GLN1;3* were significantly down-regulated following 2*Nr*EO in comparison to non-treated plants. In contrast to shoots, all *GLN1* genes except *GLN1;4*, as well as *GLN2* were down-regulated by BASTA, and the inhibitory effect was dose-dependent. Simultaneous application of BASTA and *Nr*EO severely reduced the inhibitory effect of BASTA on GS gene expression in roots, indicating the regulation on transcriptional level. Such gene expression profiles were not reflected at the protein level through GS1 and GS2 abundance in roots, which was increased in BASTA treatments. In roots, *Nr*EO-induced decrease in GS1 abundance was recorded. Simultaneous application of BASTA (B5 and B10) and *Nr*EO (2*Nr*EO and 4*Nr*EO) reduced GS1 amount in roots. Again, a strong suppression of BASTA-induced effects, following its simultaneous application with *Nr*EO, was observed 10 days after the treatment. However, the possibility that the increase in *GS* expression occurs earlier in treatments with BASTA, which is followed by the increase in GS abundance visible at 10DAT, should not be neglected. After that, the expression of GS probably decreases. 

Ammonia accumulation in response to GS inhibition is often considered to be the driver of phosphinothricin toxicity. Along with the inhibition of GS, BASTA action also leads to oxidative stress in plants, which is most probably a secondary effect of the altered metabolic pathways [42]. It has recently been suggested that glufosinate is toxic to plants, not because of ammonia accumulation or carbon assimilation inhibition, but the production of ROS, which drive the lipid peroxidation of the cell membranes and cell death [16,17]. Plants have developed mechanisms to cope with oxidative stress induced by ROS accumulation, which include enzymatic and non-enzymatic antioxidants [43]. Thus, our intention was to investigate alterations in the antioxidant system of Arabidopsis grown in vitro, induced by the application of BASTA and/or *Nr*EO. BASTA (B5 and B10) and *Nr*EO (2*Nr*EO and 4*Nr*EO), when applied independently for 10 days, altered the activity and abundance of CAT, POX and SOD in Arabidopsis shoots and roots. Induced changes were more pronounced in roots than in shoots. BASTA induced significant increases in both CAT and POX activity and abundance in roots, while SOD activity was not affected. The increase in CAT and POX activities indicated that application of glufosinate induced the increasing ROS level in Arabidopsis roots, and that enzyme activities increased correspondingly in order to eliminate excess ROS. In shoots, B10 treatment induced decrease of CAT activity and abundance, while POX activity was increased. Superoxide dismutase activity in Arabidopsis shoots was not significantly affected by BASTA treatments. While the total SOD activity in shoots and roots remains unchanged in response to applied BASTA and *Nr*EO treatments, the contribution of individual isoforms to the overall activity is variable, depending on the BASTA treatment. Mn-SOD abundance in shoots was decreased following BASTA treatment, while in roots, the abundance of this isoform increased when B5 was applied. In Arabidopsis shoots and roots B5 treatment increased the abundance of Fe-SOD isoform. It has been recently suggested that glufosinate affects the balance between ROS generation and scavenging in *Amaranthus palmeri*, and induces the increase in CAT, POX and SOD activities in an attempt to quench the nascent ROS burst [15]. SOD participates in the removal of O_2_·^−^ by conversion to H_2_O_2_ and O_2_, and its activity reflects the readiness of plants to scavenge cellular free radicals. CAT and POX can effectively remove intracellular H_2_O_2_ and translate it into H_2_O and O_2_. The increased activity and abundance of CAT and POX following BASTA treatment indicated that these two enzymes are the major responsible for ROS quenching. Similarly, activities of antioxidant enzymes (CAT, POX, SOD) increased in the leaves of maize seedlings treated by PPT [44]. Savić et al. [45] demonstrated that BASTA induced an increase in POX activity (about 42%) and appearance of new POX isoforms in *Lotus corniculatus* plants. In *Chlorella vulgaris* PPT increased CAT and SOD activity about 2.9 times, four days after application compared to control [46].

Interestingly, the effect of *Nr*EO on the activity of antioxidant enzymes in Arabidopsis shoots was opposite to those recorded for BASTA. Decrease in CAT activity in both Arabidopsis shoots and roots was followed by a decrease in CAT abundance in these organs. The same trend, which was more pronounced in roots, was observed for POX activity. These results are in accordance with previous studies, which described *Nr*EO-induced alterations in the antioxidative defense system of ragweed (*Ambrosia artemisifolia*) shoots, characterized by increased POX activity and decreased CAT and SOD activities [29]. Furthermore, reference [35] showed that nepetalactone-rich EOs decreased POX, CAT, and SOD (Fe-SOD and CuZn-SOD) activities in cress seedlings. The decrease in POX, CAT, and SOD activities in Arabidopsis roots was recorded for 2*Nr*EO treatment, which might be the result of decreased ROS formation and further studies are necessary to confirm this hypothesis. However, SOD activity was increased in Arabidopsis roots treated with higher concentrations of *Nr*EO (4*Nr*EO), which suggests that this enzyme is active in scavenging O_2_·^−^ in roots. It is well documented that components of essential oils display phytotoxic effects by generating ROS, and inducing oxidative stress [29,47,48]. Singh et al. [47] demonstrated that the content of H_2_O_2_ was increased, as well as the activity of CAT, POX and SOD, in *Cassia occidentalis* roots after *α*-pinene application. *α*-Pinene induced oxidative stress in *C. occidentalis* roots, which was visible through the disruption of membrane integrity, lipid peroxidation, H_2_O_2_ generation, proline accumulation, and increased activities of SOD, ascorbate peroxidase (APX), guaiacol peroxidase (GPX), CAT, and glutathione reductase (GR). 

Simultaneous application of *Nr*EO and BASTA, two agents inducing opposite effects on the Arabidopsis antioxidant system, especially in roots, mitigated the effects of the two agents applied independently. This was evident for CAT and POX activities, which were significantly lower on *Nr*EO+B5 treatments (both 2*Nr*EO and 4*Nr*EO) than on B5 treatments, and higher than on *Nr*EO treatments. However, simultaneous application of BASTA and *Nr*EO significantly increased SOD activity in Arabidopsis roots, which suggests that this enzyme is more active in scavenging O_2_^−^ than on BASTA treatments. *Nr*EO preserves the GS activity and maintains subtoxic ammonia levels in Arabidopsis leaves during simultaneous BASTA and *Nr*EO application [4], and it most likely acts by inducing SOD activity in roots and thus more efficient ROS scavenging, which contributes to mitigating the phytotoxic effects of BASTA. Our further work, involving sophisticated methods and tools (e.g., ROS tissue localization, analysis of other antioxidant enzymes activity at the protein and gene expression level, metabolomics, tracing PPT uptake and metabolism in Arabidopsis), will help us to comprehensively explain the phenomenon of antagonistic interaction between PPT and *Nr*EO. In summary, *Nr*EO has elicited considerable agronomic interest, and future efforts directed towards unraveling the mechanism by which these two agents interact will contribute to the possible utilization of *Nr*EO as an eco-friendly bioherbicide and an agent for mitigation of the effects of PPT residues in the soil on non-target plants. 

## 4. Materials and Methods

### 4.1. Chemicals and Reagents

Commercial herbicide BASTA^®^ (containing 150 g L^−1^ of active ingredient glufosinate-ammonium) was purchased from BASF SE (Germany). Solvents for HPLC–MS analyses (acetonitrile, acetic and formic acids) were of LC–MS grade (Fisher Scientific, Loughborough, UK). Methanol for metabolites extraction and preparation of EO dilutions (HPLC grade) was purchased from AppliChem (Cheshire, CT, USA). Ultrapure water was generated by deionization (Millipore, Billerica, MA, USA). Standards for sugars and organic acids determination were purchased from Sigma-Aldrich (Sigma Co., St. Louis, MO, USA).

### 4.2. Preparation of Essential Oils

*Nepeta rtanjensis* Diklić & Milojević plants were cultivated at the experimental field of the Institute for Biological Research “Siniša Stanković”—National Institute of the Republic of Serbia, University of Belgrade, Serbia. Aerial parts of flowering plants were harvested, air-dried and used for the isolation of EO by hydrodistillation, as previously described by Skorić et al. [31]. To obtain 2% (*v*/*v*) and 4% (*v*/*v*) final EO concentrations, *N. rtanjensis* EO (*Nr*EO) was diluted in 99.8% methanol.

### 4.3. Joint Effect of BASTA via Roots and NrEO on A. thaliana Plants—In Vitro Phytotoxic Assay

Seeds of *Arabidopsis thaliana* (L.) Heynh, accession Col−8 (N60000), were obtained from the Nottingham Arabidopsis Stock Centre (http://arabidopsis.info). Arabidopsis seeds were surface sterilized in 20% solution of commercial bleach in ethanol (0.8% active chlorine) for 1 min and subsequently rinsed 5 times in sterile deionized water. Seeds were transferred into 9 cm Petri dishes containing 20 mL of basal medium (BM), a modified Murashige and Skoog [49] medium, supplemented with half-strength macro elements, 20 g L^−1^ sucrose and 7 g L^−1^ agar. The pH of the culture medium was adjusted to 5.8 before sterilization by autoclaving at 114 °C for 25 min. Seeds were stratified for 3 days at 4 °C in the dark. Petri dishes were subsequently transferred to a growth chamber at 25 ± 2 °C, under 16/8 h light/dark regime with a photon flux density of 70 µmol m^−2^ s^−1^. After 11 days, Arabidopsis seedlings were transferred into 250 mL Erlenmeyer flasks containing 100 mL BM, and grown under the same light regime. 

After two weeks, five well-developed seedlings were placed in 350 mL glass jars containing 75 mL BM, with or without dissolved BASTA at final concentrations of active ingredient (glufosinate ammonium) 5 mg L^−1^ (B5) or 10 mg L^−1^ (B10). Fifty µL of diluted *Nr*EO (2 or 4% solution in methanol) was applied to the filter paper (2.5 × 2.5 cm) which was subsequently rolled up, placed on a sterilized metal holder, and stuck into the culture medium with care not to contact the plants. Volatile components of the EO evaporated from the filter paper surface into the atmosphere of glass jar, where Arabidopsis seedlings were cultivated. For the negative control, filter paper was moistened with 50 µL of 99.8% methanol. Thus, 9 experimental groups were prepared: (1) non-treated plants (nt, plants exposed to methanol only), (2) methanol + BASTA 5 mg L^−1^ (B5), (3) methanol + BASTA 10 mg L^−1^ (B10), (4) 2% EO (2*Nr*EO), (5) 4% EO (4*Nr*EO), (6) 2% EO + 5 mg L^−1^ BASTA (2*Nr*EO+B5), (7) 4% EO + 5 mg L^−1^ BASTA (4*Nr*EO+B5), (8) 2% EO + 10 mg L^−1^ BASTA (2*Nr*EO+B10), and (9) 4% EO + 10 mg L^−1^ BASTA (4*Nr*EO+B10). All glass jars were closed with polycarbonate caps, and placed in a growth chamber at 25 ± 2 °C, under 16/8 h light/dark regime, with a photon flux density of 70 µmol m^−2^ s^−1^. After ten days, *A. thaliana* shoots and roots were separately pooled, weighed, frozen in liquid nitrogen (LN), and stored at −80 °C until further use. Plant material for protein analysis (Native-PAGE electrophoresis and GS activity determination) was weighed and immediately used in assays. The results were obtained using three biological replicates. Each biological replicate represented a pool of tissues collected from five plants grown in the single glass jar.

### 4.4. Determination of Volatile Organic Compounds (VOCs) Concentration in the Atmosphere of Culture Vessels

Concentrations of *Nr*EO VOCs in the atmosphere of glass vessels used for Arabidopsis cultivation were measured by Proton Transfer Reaction Mass Spectrometer (Standard PTR-quad-MS, Ionicon Analytik, GmbH, Innsbruck, Austria). Concentrations of VOCs were recorded at the beginning of the experiment, and after 10 days of treatments. Analysis was targeted towards compounds displaying molecular ions [M+1]^+^ at *m*/*z* 137, 153, 167, and 205 (0.1 s dwell time), and were obtained in 1.8 s cycles. Drift tube parameters included: pressure 2.11–2.13 mbar; temperature 60 °C; voltage 600 V; E/N parameter 145 Td; and reaction time 90 µs. The count rate of H_3_O^+^/H_2_O was 2.1–14.1% of the count rate of H_3_O^+^ ions, which was in the range 2.1 × 10^6^−6.1 × 10^6^ counts s^−1^. The calibration was done according to Taipale et al. [50], using TO−15 Supelco gas mixture (*m*/*z* 57, *m/z* 79, *m*/*z* 93, *m*/*z* 107, and *m*/*z* 121), diluted with ASGU 370-p HORIBA system zero air to five concentrations ranging from 0.5 to 100 parts-per-billion (ppb). For the calculation of transmission coefficients above *m*/*z* 170, the logarithmic extrapolation was used. Calculated normalized sensitivities were 2.23, 1.77, 1.52, and 1.33 npcs ppb^−1^ for *m*/z 137, 153, 167, and 205, respectively. Nepetalactones (*cis*,*trans*- and *trans*,*cis*-nepetalactone) showed [M+1]^+^ at *m*/*z* 167. Compounds with [M+1]^+^ at *m*/*z* 137 most likely corresponded to monoterpenoids *α*- and *β*-pinene, while *α*-campholenal, neral and geranial showed [M+1]^+^ at *m*/*z* 153. Sesquiterpenoids showing protonated mass in PTR analysis of [M+1]^+^ at *m*/*z* 205, corresponding to *γ*-cadinene, *δ*-cadinene, *cis*- and *trans*-caryophyllene, and *α*-humulene, were also analyzed. Measurements were performed in triplicate for each of the treatments. Values are presented as parts-per-billion-volume (ppbV).

### 4.5. Metabolic Profiling of Sugars, Organic Acids and Chlorophyll

HPLC analyses of soluble sugars (sucrose-Suc, fructose-Fru, and glucose-Glu) and organic acids (oxalic, succinic, malic, fumaric and isocitric acid) were performed as described earlier by Dmitrović et al. [4]. Values recorded for soluble sugars are calculated as mg 100 mg^−1^ FW, and for organic acids are µg mg^−1^ FW. Chlorophylls (Chl) were extracted from Arabidopsis shoots following the modified method of Porra et al. [51] and described in detail by Dmitrović et al. [4]. The content of Chl a, Chl b, and Chl a+b is calculated as μg g^−1^ FW.

### 4.6. Quantitative RT-PCR Analysis of GS-Coding Genes Expression

RNA extraction and the subsequent cDNA preparation were performed as described in Dmitrović et al. [4]. Quantitative RT-PCR (qRT-PCR) was conducted by using primers for *GLN1;1*, *GLN1;2*, *GLN1;3*, *GLN1;4* and *GLN2*, as described before [4]. *18S rRNA* was used as a reference gene. Relative quantification of gene expression was performed, as described by Livak and Schmittgen [52]. The results are presented as log2 fold change in expression compared to control treatments.

### 4.7. Determination of GS Activity and Ammonium Content

Protein extraction, native polyacrylamide gel electrophoresis (Native-PAGE) separation and activity staining of GS (EC 6.3.1.2) isoforms was performed as described by Dragićević et al. [53]. Total GS activity in protein extracts was assayed as described by Nikolić et al. [5]. Total ammonium content in leaves was determined by the phenol-hypochlorite method described in detail by Dmitrović at al. [4].

### 4.8. Determination of CAT, POX and SOD Activities

Antioxidant enzymes extraction and the activity determination were carried out as described by Dmitrović at al. [29]. Detailed procedures for Native-PAGE used for CAT (EC 1.11.1.6), POX (EC 1.11. 1.7) and SOD (EC 1.15.1.1) activity determination were described by Dmitrović at al. [29], with some modifications. POX activity was visualized after incubation of gels in staining solution containing 10% 4-chloro-α-naphthol and 0.03% H_2_O_2_ in 50 mM K-phosphate buffer pH 6.5.

### 4.9. SDS-PAGE Electrophoresis and Immunoblotting

Sodium dodecyl sulfate polyacrylamide gel electrophoresis (SDS-PAGE) and subsequent transfer of proteins onto nitrocellulose membranes were performed, as reported by Dmitrović et al. [29]. Immunoblot analysis was conducted using rabbit polyclonal antibodies GLN1 GLN2 (AS08295), CAT (AS09501), MnSOD (AS09524), FeSOD (AS06125), or sheep polyclonal antibodies for POX (AS09548), all purchased from Agrisera, Sweden. Primary antibodies were used in dilutions of 1:2500 (GS), 1:1000 (CAT), 1:2000 (POX), 1:1000 (MnSOD), and 1:1000 (FeSOD). In the case of GS, CAT, Mn-SOD and Fe-SOD goat anti-rabbit IgG-peroxidase conjugated (A0545, Sigma-Aldrich, USA) secondary antibodies were used, while goat anti-mouse IgG HRP conjugated (AS111772, Agrisera, Sweden) antibodies were applied for POX. Goat anti-rabbit IgG-peroxidase antibodies were diluted to 1:20,000 (*v*:*v*), and goat anti-mouse IgG HRP were diluted to 1:5000 (*v*:*v*). After intensive washing in PBS buffer, protein signals were visualized using an enhanced chemiluminescence detection system (ECL). Densitometric analysis of band intensities was performed using ImageJ 1.32j software (W. Rasband, National Institute of Health, Bethesda, MD, USA). The obtained signal intensities were normalized to the highest value, and the results are presented as the relative abundances.

To confirm equal loading in immunoblots, membranes were incubated for 2 h in primary Actin-11 antibodies (Anti Mouse monoclonal IgG2b lyophilized, AS10702; Agrisera Antibodies, Sweden) diluted to 1:1000 (*v:v*) and goat anti-mouse IgG horse radish peroxidase conjugated secondary antibodies (AS11 1772, Agrisera Antibodies, Sweden) diluted to 1:50,000 (*v*:*v*) using the procedure described above. The obtained signal intensities for GS were normalized to the actin values, and the obtained results were normalized to the highest value and presented as relative abundances.

### 4.10. Statistical Analyses

Statistical analyses were performed using R [54]. The data were subjected to “sequential” sum of squares (type I SS ANOVA) factorial ANOVA by testing the main effect of BASTA, followed by the main effect of *Nr*EO after the main effect of BASTA, followed by the interaction effect *Nr*EO+BASTA after the main effects. For the analyses of qRT-PCR data log2-fold change was used as a response variable in factorial ANOVA. For all other models, the homoscedasticity and normality of the residuals were checked graphically, and if these assumptions were violated, the data was transformed prior to the statistical analyses using Box-Cox power transformation [55] incorporated in the R library MASS [56]. ANOVA was followed by Tukey’s post-hoc test at the *p* < 0.05 significance level using the R library emmeans [57].

## Figures and Tables

**Figure 1 plants-10-00142-f001:**
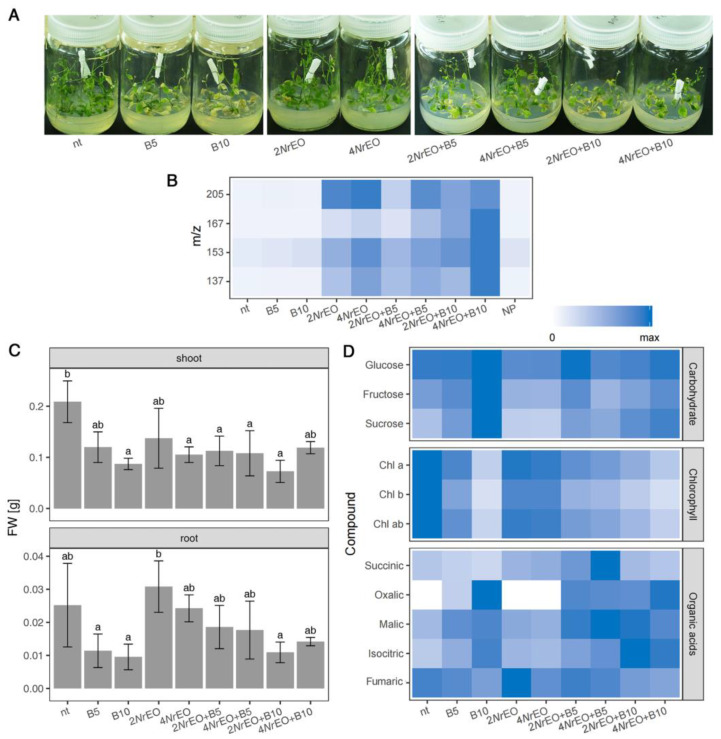
The effects of herbicide BASTA (B5 and B10, with 5 and 10 mg L^−1^ glufosinate ammonium, respectively), and *N. rtanjensis* essential oil volatiles (2*Nr*EO and 4*Nr*EO with 2% and 4% *Nr*EO, respectively) and their combined application on Arabidopsis plants. (**A**) *In vitro* grown Arabidopsis exposed to different combinations of BASTA and/or *Nr*EO, as well as non-treated (nt) plants. BASTA was supplied through culture medium, while *Nr*EO volatiles were present in the atmosphere of the glass jars. (**B**) Values of the relative abundances (ppbV, parts-per-billion-volume) of organic volatiles present in the atmosphere of the glass vessels after 10 days of the treatment, revealed by the PTR-MS measurements, are presented relatively as a heat-map: [H+1]^−^ at *m*/*z* 167—*cis*,*trans*-nepetalactone + *trans*,*cis*-nepetalactone; [M+1]^−^ at *m*/*z* 137 and *m*/*z* 153—total monoterpenoids (*m*/*z* 137—α- and β-pinene, *m*/*z* 153—α-campholenal, neral and geranial); [H+1]^−^ at *m*/*z* 205—total sesquiterpenoids (γ- and δ- cadinene, *cis*- and *trans*-caryophyllene, and α-humulene). (**C**) Arabidopsis shoot fresh weight (FW) and root FW. Values are presented as means ± SD. Significant differences according to Tukey’s HSD post-hoc test at *p* < 0.05 are indicated with a compact letter display. (**D**) Content of chlorophylls (Chl a, Chl b, Chl a+b), organic acids (succinic, oxalic, malic, isocitric and fumaric acid) and soluble sugars (glucose, fructose and sucrose) in shoots. Maximal values on the colour scales represent maximal values recorded for each parameter. Abbreviations: nt- non treated plants; NP- culture vessels containing no plants; B- BASTA; *Nr*EO- *Nepeta rtanjensis* essential oil.

**Figure 2 plants-10-00142-f002:**
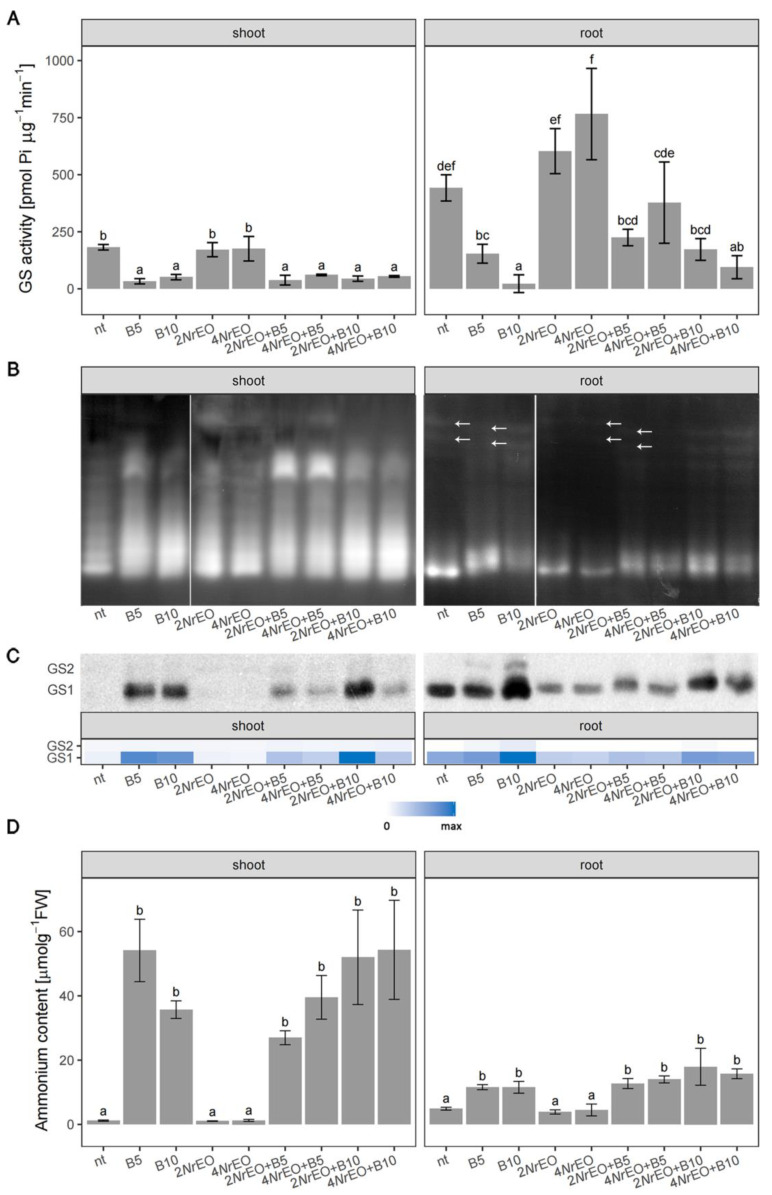
Dose-dependent effects of BASTA (B5 and B10, with 5 and 10 mg L^−1^, respectively), *N. rtanjensis* essential oil (2*Nr*EO and 4*Nr*EO with 2% and 4% *Nr*EO, respectively), and their mixtures (*Nr*EO + BASTA) on *in vitro* grown Arabidopsis plants. (**A**) Total GS activity in shoots and roots. Values are presented as means ± SD, and significant differences according to Tukey’s HSD *post-hoc* test at *p* < 0.05 are indicated with a compact letter display. (**B**) Distribution of GS isoforms activity from shoot and root extracts, stained after native PAGE (50 µg total protein per well). Arrows indicate a mobility shift of GS bands. (**C**) Immunoblots conducted with specific GS antibodies. Heat-maps show relative abundances of GS proteins. Maximal values on the color scales represent maximal values recorded for each immuno blot, independently. (**D**) Ammonium content in shoots and roots. Values are presented as means ± SD. Significant differences according to Tukey’s HSD post-hoc test at *p* < 0.05 are indicated with a compact letter display.

**Figure 3 plants-10-00142-f003:**
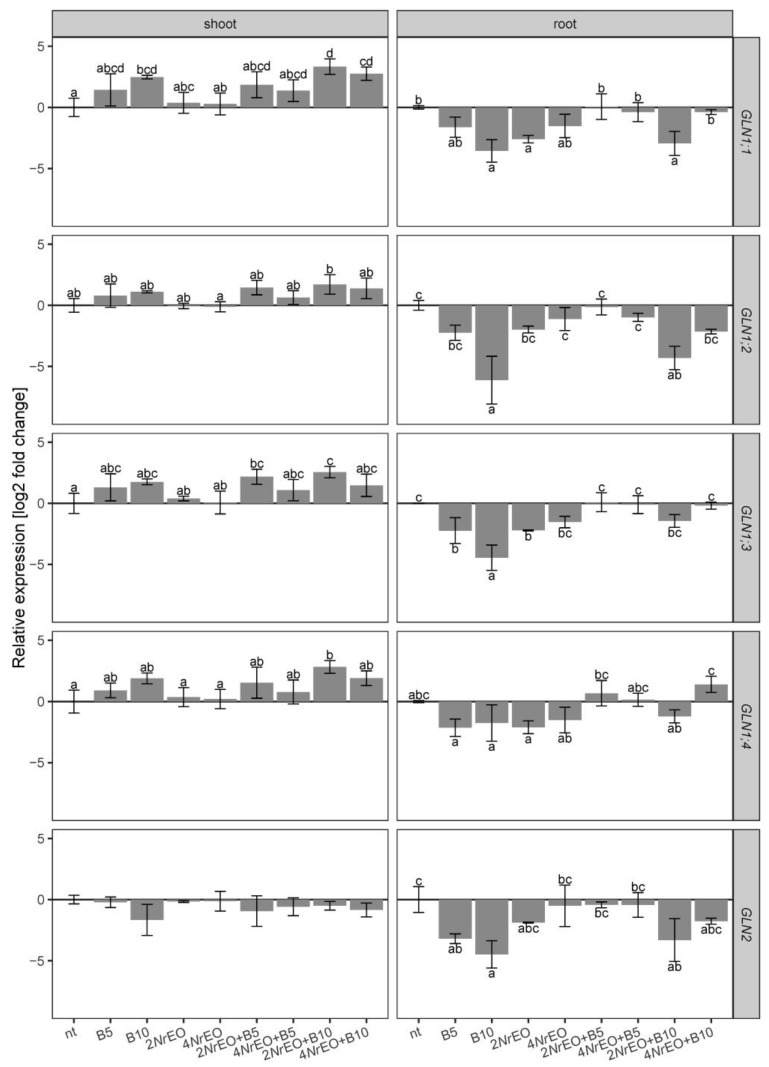
The effects of BASTA applied at two concentrations B5 (5 mg L^−1^) and B10 (10 mg L^−1^), *N. rtanjensis* essential oil applied at two concentrations 2% (2*Nr*EO) and 4% (4*Nr*EO), and of their combinations on the relative expression of GS coding genes in Arabidopsis shoots and roots, measured 10 days after the beginning of the treatment, grown *in vitro*. Values are presented as means ± SD. Letters above the bars denote significant differences according to Tukey’s HSD post-hoc test at *p* < 0.05. The expression of *GLN1;5* was not detected.

**Figure 4 plants-10-00142-f004:**
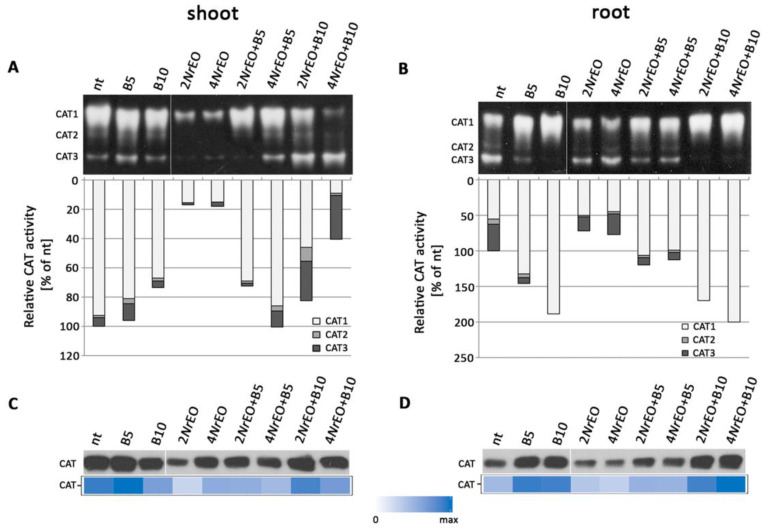
Catalases (CAT) activity in Arabidopsis grown *in vitro*, measured 10 days after treatment with BASTA applied at two concentrations B5 (5 mg L^−1^) and B10 (10 mg L^−1^), *N. rtanjensis* essential oil applied at two concentrations 2% (2*Nr*EO) and 4% (4*Nr*EO), and of their combinations. (**A**,**B**) CAT activity stain after native-PAGE separation (5 µg per lane of total protein), in shoots and roots, respectively. The detected activities were measured densitometrically and presented as relative CAT activity (% of control) in comparison to non-treated plants (nt). (**C**,**D**) CAT immunoblot (20 µg of soluble proteins per line) in shoots and roots, respectively. Heat-maps show relative abundances of CAT proteins. Maximal values on the color scales represent maximal values recorded for each immunoblot.

**Figure 5 plants-10-00142-f005:**
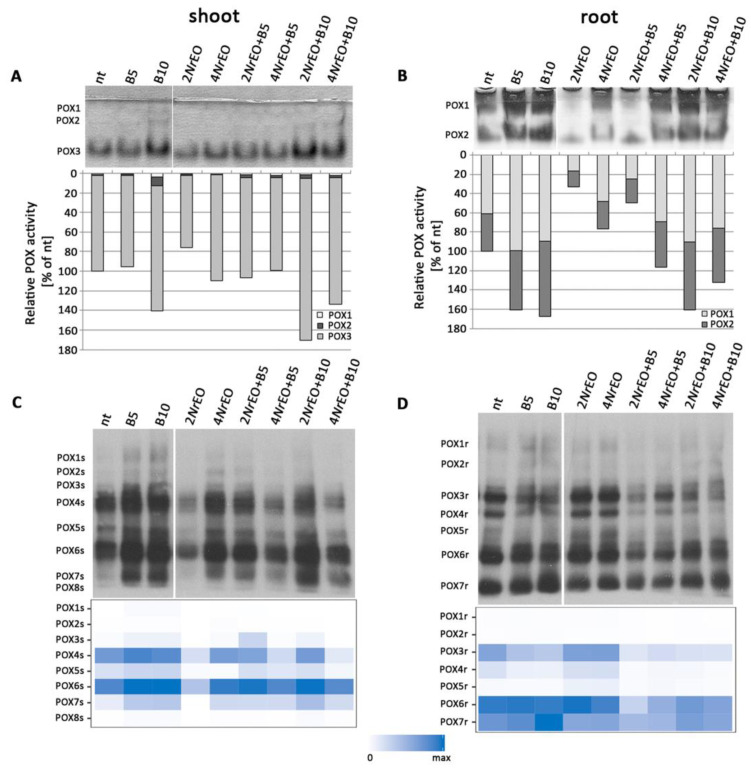
Peroxidases (POX) activity in Arabidopsis grown *in vitro*, measured 10 days after treatment with BASTA applied at two concentrations B5 (5 mg L^−1^) and B10 (10 mg L^−1^), *N. rtanjensis* essential oil applied at two concentrations 2% (2*Nr*EO) and 4% (4*Nr*EO), and of their combinations. Total soluble proteins were extracted from shoots and roots and loaded (20 µg per lane) on 10% gel, separated by native-PAGE and assayed for POX activity. (**A**,**B**) The detected activities were measured densitometrically and presented as relative POX activity (% of control) in comparison to non-treated plants (nt). (**C**,**D**) For SDS-PAGE 20 µg of soluble proteins per line from shoots and roots were loaded on 10% gel, transferred to nitrocellulose membrane and immunoblotted using primary antibodies antisheep horseradish POX. Corresponding heat-maps show relative abundances of POX proteins. Maximal values on the colour scales represent maximal values recorded for each immunoblot, independently. Abbreviations: s—shoot, r—root.

**Figure 6 plants-10-00142-f006:**
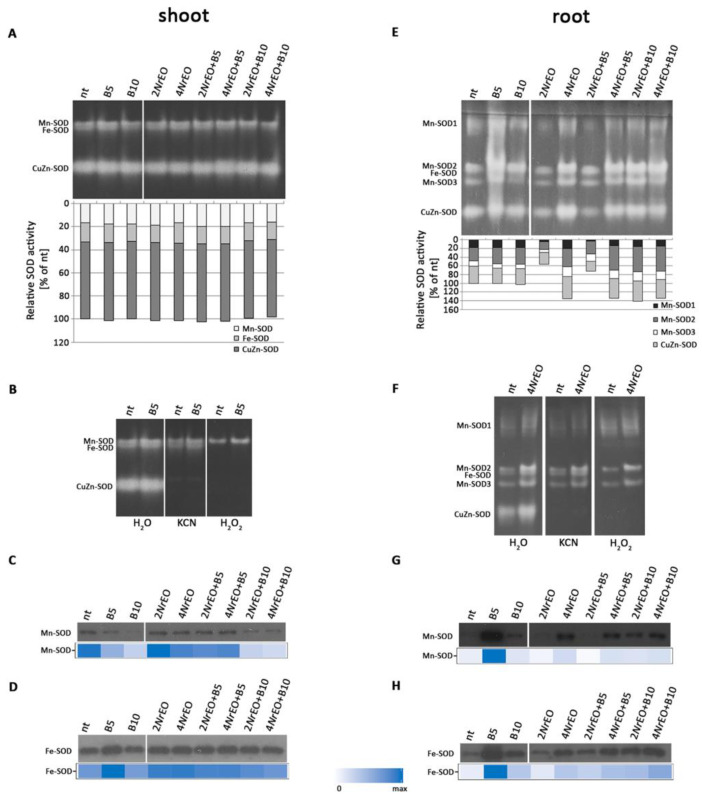
Superoxide dismutase (SOD) activity in Arabidopsis shoots and roots grown *in vitro*, measured 10 days after treatment with BASTA applied at two concentrations B5 (5 mg L^−1^) and B10 (10 mg L^−1^), *N. rtanjensis* essential oil applied at two concentrations 2% (2*Nr*EO) and 4% (4*Nr*EO), and of their combinations. Total soluble proteins were extracted from shoots and roots and loaded (5 µg per lane) on 10% gel, separated by native-PAGE and assayed for SOD activity. (**A**,**E**) The detected activities were measured densitometrically and presented as relative SOD activity (% of control) in comparison to non-treated plants (nt). (**B**,**F**) Identification of SOD isoforms based on differential sensitivity to hydrogen peroxide and potassium cyanide. (**C**–**H**) For SDS-PAGE 20 µg of soluble proteins per line from shoots/roots were loaded on 10% gels, transferred to nitrocellulose membranes and immunoblotted using primary antibodies antirabbit MnSOD (**C**,**G**) and antirabbit chloroplastic FeSOD (**D**,**H**). Corresponding heat-maps show relative abundances of SOD proteins. Maximal values on the color scales represent maximal values recorded for each immunoblot, independently.

## Data Availability

The data presented in this study are available on request from the corresponding authors.

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
