# Peer review of "Antagonistic Interaction between Phosphinothricin and Nepeta rtanjensis Essential Oil Affected Ammonium Metabolism and Antioxidant Defense of Arabidopsis Grown In Vitro"

_plants, 2021, doi:10.3390/plants10010142_

Round 1
Reviewer 1 Report
This work demonstrates that the interaction of Nepeta rtanjensis essential oil (NrEO) and the herbicide phosphinothricin (PPT) result in the mitigation of PPT-induced ammonium toxicity and provides a more efficient antioxidant defense of plants. The topic is important from an agronomic point of view, due to the environmental damage incited by herbicides. However, the conclusion that NrEO confers a more efficient antioxidant plant defense (also to PPT) by elevating SOD activity should be elaborated a bit in the Discussion. Namely, elevated SOD activities suggest that the main cause of PPT-induced oxidative stress is the accumulation of the ROS superoxide (O2·-). Superoxide levels (e.g. by simple NBT tissue staining) assayed in plant tissues (shoots and roots) exposed to PPT and/or NrEO could reveal if this hypothesis is right. Also, testing whether in planta superoxide generation by e.g. riboflavin/methionine or xanthine-xanthine oxidase could amplify PPT-induced oxidative stress could lead to a similar answer. I do not insist that the authors should conduct these experiments but they should mention these experimental approaches as possible tools for proving the hypothesis (role of superoxide in PPT-induced oxidative stress).
Comments:
1/ In Line 99 and Fig. 1 relative abundance (ppbV) means "pars per billion volume"? Please clarify.
2/ It would be helpful for the reader to indicate MW (kDa) of PAGE bands in Fig. 2B
For additional minor comments see the attached MS file with my corrections.

Author Response
Reviewer #1
This work demonstrates that the interaction of Nepeta rtanjensis essential oil (NrEO) and the herbicide phosphinothricin (PPT) result in the mitigation of PPT-induced ammonium toxicity and provides a more efficient antioxidant defense of plants. The topic is important from an agronomic point of view, due to the environmental damage incited by herbicides. However, the conclusion that NrEO confers a more efficient antioxidant plant defense (also to PPT) by elevating SOD activity should be elaborated a bit in the Discussion. Namely, elevated SOD activities suggest that the main cause of PPT-induced oxidative stress is the accumulation of the ROS superoxide (O2·-). Superoxide levels (e.g. by simple NBT tissue staining) assayed in plant tissues (shoots and roots) exposed to PPT and/or NrEO could reveal if this hypothesis is right. Also, testing whether in planta superoxide generation by e.g. riboflavin/methionine or xanthine-xanthine oxidase could amplify PPT-induced oxidative stress could lead to a similar answer. I do not insist that the authors should conduct these experiments but they should mention these experimental approaches as possible tools for proving the hypothesis (role of superoxide in PPT-induced oxidative stress).
Answer: We are grateful to the Reviewer for helpful suggestions and comments. Our intention was to avoid speculations and to draw conclusions based only on the obtained results. Although there are some indications, we were not able to conclude that the accumulation of O2·- is the main cause of PPT-induced oxidative stress. PPT alone is not inducing the SOD activity in shoots and roots. Only when PPT and NrEO are applied together the increase in SOD activity in roots can be observed. Actually, our further work will be conducted towards collecting more information about the modes of PPT and NrEO action at the level of antioxidant system, in a temporal manner. Our intention is to adopt sophisticated tools and methods (ROS tissue localization using fluorescent dyes, analysis of other antioxidant enzymes activity at the protein and gene level, metabolomics, etc), in order to provide deeper insight into this interesting phenomenon. As the experiments presented within this MS are performed in 2018, and we have no plant material left, at this point we are not able to perform the analyses that Reviewer optionally suggested. We introduced more explanations about our further studies into the last paragraph of the Discussion section.
Comments:
1/ In Line 99 and Fig. 1 relative abundance (ppbV) means "pars per billion volume"? Please clarify.
Answer: Common unit of indoor air measurements- ppbV, means parts-per-billion-volume. We have introduced the explanation into the Figure 1 legend, and also in the M&M section, subsection 4.4.
2/ It would be helpful for the reader to indicate MW (kDa) of PAGE bands in Fig. 2B
Answer: Fig. 2B represents the NATIVE-Page zymograms of GS activity, where multiple activity bands are visible. GSs are decamers, and Arabidopsis genome encodes five cytosolic (GS1) proteins labeled as GLN1;1 to GLN1;5 and one chloroplastic (GS2) isoform, GLN2;0. According to our previous study, each active GS isoform in Arabidopsis consist of varying proportions of GLN1;1 to GLN1;3 subunits, which form different homo- or heterodecamers (for more details, please see our previous paper Dragićević et al., 2014). As Figure 2B represents NATIVE-Page zymograms, it is not possible to include MW of bands. However, MWs of GS1 and GS2 isoforms can be calculated from SDS-Page zymograms, which are not presented in Figure 2. We performed SDS-Page prior to Western Blotting, and it was possible to calculate the MWs of GS1 and GS2, which were 40 kDa and 44 kDa, respectively. This information is already included into the Results section, subsection 2.2.
For additional minor comments see the attached MS file with my corrections.
Answer: We have accepted the majority of suggestions and comments and introduced changes into the MS.
Reviewer 2 Report
This quite deep study attempts to explain an antagonistic effect of phosphinothricin (PPT) and essential oil of Nepeta rtanjensis applied in in vitro model to Arabidopsis thaliana. The PPT is applied to media to investigate the interaction while the PPT is absorbed via roots although this herbicide is routinely applied as a foliar spray. The authors based the experimental design on the fact that herbicide reaches the soil and remains 1-25 days before being degraded, thus can be absorbed by non-target plants via roots. In the previous study, the authors proved the antagonistic effect of PPT and Nepeta rtanjesis essential oil when sprayed in combination on plant foliage. They suggested that the application of the essential oil might mitigate the negative effect of PPT residues in soil. Therefore I assume this study might be important and definitely is novel. The paper can be accepted for publication, I only have few following comments:
- I suggest to consider setting the max value of Y axis in Fig 1C, chart for root to 0.1 or even 0.05 for better visualization of differences.
- The treatment “NP” in Figure 1B is not explained.
- Please add information on final concentration of the active compound (glufosinate) in chapter 4.3, row 594. The concentration used in experimental model should be justified and confronted with theoretical or real (if any data available) concentrations of glufosinate and its residues in soil.
- Justify, why the equivalent control treatment to B5 and B10 treatment was not applied. The control treatment with MeOH only was equivalent to treatments where essential oil was used. MeOH in air itself might have some effect, though minor, thus it is not suitable as a negative control to solely applied herbicide in my opinion.
- It is a pity the authors did not monitor the PPT degradation kinetics in growth media within 10 experimental days. Contrarily the volatile organic compounds concentration was recorded in cultivation vessels after 10 days of cultivation.
Author Response
Reviewer #2
This quite deep study attempts to explain an antagonistic effect of phosphinothricin (PPT) and essential oil of Nepeta rtanjensis applied in in vitro model to Arabidopsis thaliana. The PPT is applied to media to investigate the interaction while the PPT is absorbed via roots although this herbicide is routinely applied as a foliar spray. The authors based the experimental design on the fact that herbicide reaches the soil and remains 1-25 days before being degraded, thus can be absorbed by non-target plants via roots. In the previous study, the authors proved the antagonistic effect of PPT and Nepeta rtanjesis essential oil when sprayed in combination on plant foliage. They suggested that the application of the essential oil might mitigate the negative effect of PPT residues in soil. Therefore I assume this study might be important and definitely is novel. The paper can be accepted for publication, I only have few following comments:
- I suggest to consider setting the max value of Y axis in Fig 1C, chart for root to 0.1 or even 0.05 for better visualization of differences.
Answer: We appreciate the Reviewer’s suggestion, and we agree that setting the Y axis scale to 0.05 would result in more visible changes in FW of Arabidopsis roots between the treatments. We now provide the revised Figure 1.
- The treatment “NP” in Figure 1B is not explained.
Answer: According to the Reviever’s suggestions, we introduced the explanations for the “NP” abbreviation into the Figure 1 legend, together with the explanation of other abbreviations in this figure. “NP” stands for “no plant”, referring to the atmosphere of culture vessels containing no plants.
- Please add information on final concentration of the active compound (glufosinate) in chapter 4.3, row 594. The concentration used in experimental model should be justified and confronted with theoretical or real (if any data available) concentrations of glufosinate and its residues in soil.
Answer: The final concentrations of active ingredient (glufosinate-ammonium) in culture medium were 5 mg l-1 and 10 mg l-1. We introduced more precise explanation into the M&M section, subsection 4.3. The concentrations applied in experiments are selected based on our previous study (Dmitrović et al., 2019) and many other studies with Arabidopsis, and represent sub-lethal concentrations of herbicide inducing phytotoxicity in Arabidopsis. Additionally, we included more detailed description of the commercial BASTA formulation used in the experiments (section M&M, subsection 4.1.).
- Justify, why the equivalent control treatment to B5 and B10 treatment was not applied. The control treatment with MeOH only was equivalent to treatments where essential oil was used. MeOH in air itself might have some effect, though minor, thus it is not suitable as a negative control to solely applied herbicide in my opinion.
Answer: We did some preliminary experiments with two negative controls: one with MeOH and one without any treatment (data not presented within the MS). As we didn’t record any significant differences in parameters analysed within the present study between the two treatments, we decided to continue our study using only MeOH as negative control. In order to be more precise, we now explained in M&M section (subsection 4.3.) that plants on B5 and B10 treatments were also exposed to MeOH. Thus, MeOH treatment could be used as a control for all the treatments in the experiment, which enabled us to adopt powerful statistics for the interpretation of the results.
- It is a pity the authors did not monitor the PPT degradation kinetics in growth media within 10 experimental days. Contrarily the volatile organic compounds concentration was recorded in cultivation vessels after 10 days of cultivation.
Answer: We appreciate the suggestion of Reviewer 1, and we agree that this information would be very interesting and explanatory. However, due to technical difficulties, we were not able to trace the level of PPT in culture media and in plants. One of the aims of our further work is to design and validate LC/MS analytical methods for identification and quantification of PPT and its residues in biological material. One of the hypotheses of the present study is that PPT present in in vitro culture medium is not degraded by the microorganisms as this medium is sterile. Thus, the decrease of PPT level in culture medium is exclusively the result of its uptake by plants. Components of the NrEO, on the other hand, are volatile, and may leak from culture vessels into the “ex vitro” atmosphere, which results in the reduction of their content in the atmosphere of glass vessels during 10 days of experiment. This is why we traced the amount of major VOCs in the atmosphere of culture vessels at the beginning and in the end of the experiments.
Reviewer 3 Report
The concept of this review article is interesting. The paper is written clearly and understandably. However, there are some minor errors in the text that should be corrected. Some specific comments about the content are listed below. I have used the page numbers that are printed in the right margin of the document.
- Page2L48 - change ‘free radicals’ to ‘ROS’.
- There is no research hypothesis. The research hypothesis should take the form of a statement (not a question or guess). The hypothesis should always explain what you expect to happen. Rejecting the null hypothesis and accepting the alternative hypothesis is the basis for building a good research study. Please, correct accordingly.
- Page2L70-72 - The authors wrote: ‘Thus, one of the aims of the present study was to investigate the herbicidal effect of PPT present in the root surrounding, by tracing its effects on the Arabidopsis physiology and biochemistry.’ I don’t think that this sentence is necessary, it is better to focus on the formulation of the research hypothesis, possibly auxiliary hypotheses, in the further part of Introduction.
- Do deviations from the means in the figures mean SD or SE? Please, add this information.
- Page5L213- please give the abbreviation for VOCs when it appears for the first time in the text.
- I suggest to remove ‘plants’ from the phrase ‘Arabidopsis plants’ – potential readers know that they are plants.
- Page17L657-658 – please, provide the EC numbers of the enzymes.
Author Response
Reviewer #3
The concept of this review article is interesting. The paper is written clearly and understandably. However, there are some minor errors in the text that should be corrected. Some specific comments about the content are listed below. I have used the page numbers that are printed in the right margin of the document.
- Page2L48 - change ‘free radicals’ to ‘ROS’.
Answer: The term “free radiclas” is changed into “ROS”, as suggested by the Reviewer.
- There is no research hypothesis. The research hypothesis should take the form of a statement (not a question or guess). The hypothesis should always explain what you expect to happen. Rejecting the null hypothesis and accepting the alternative hypothesis is the basis for building a good research study. Please, correct accordingly.
Answer: As suggested by the Reviewer, we introduced changes into the last paragraph of the Introduction section, and defined our hypotheses.
- Page2L70-72 - The authors wrote: ‘Thus, one of the aims of the present study was to investigate the herbicidal effect of PPT present in the root surrounding, by tracing its effects on the Arabidopsis physiology and biochemistry.’ I don’t think that this sentence is necessary, it is better to focus on the formulation of the research hypothesis, possibly auxiliary hypotheses, in the further part of Introduction.
Answer: As suggested, we deleted the sentence in question, and introduced changes into the last paragraph of the Introduction section, which now focuses on the research hypotheses formulation.
- Do deviations from the means in the figures mean SD or SE? Please, add this information.
Answer: We introduced standard deviations (SDs) on figures. We included the explanations into the legends of Figures 1, 2 and 3.
- Page5L213- please give the abbreviation for VOCs when it appears for the first time in the text.
Answer: As suggested by the Reviewer, full term (Volatile organic compounds) and abbreviation (VOCs) are given when first mentioned in the text (Page 5).
- I suggest to remove ‘plants’ from the phrase ‘Arabidopsis plants’ – potential readers know that they are plants.
Answer: We removed “plants” from the phrase ‘Arabidopsis plants’ throughout the MS, as suggested. The same was done in the Title, which is now “Antagonistic interaction between phosphinothricin and Nepeta rtanjensis essential oil affected ammonium metabolism and antioxidant defense of Arabidopsis grown in vitro”.
- Page17L657-658 – please, provide the EC numbers of the enzymes.
Answer: As suggested by the Reviewer, EC numbers of GS, CAT, POX and SOD are introduced into the M&M section, subsections 4.7 and 4.8.